# Advancement in Agriculture Approaches with Agrivoltaics Natural Cooling in Large Scale Solar PV Farms

**Noor Fadzlinda Othman** [1,2,*], **Mohammad Effendy Ya'acob** [2,3,*], **Li Lu** [4], **Ahmad Hakiim Jamaluddin** [5], **Ahmad Suhaizi Mat Su** [1], **Hashim Hizam** [4], **Rosnah Shamsudin** [3] and **Juju Nakasha Jaafar** [6]

1   Department of Agriculture Technology, Faculty of Agriculture, Universiti Putra Malaysia, Serdang 43400, Selangor, Malaysia; asuhaizi@upm.edu.my
2   Hybrid Agrivoltaic System Showcase (HAVs), Faculty of Engineering, Universiti Putra Malaysia, Serdang 43400, Selangor, Malaysia
3   Department of Process and Food Engineering, Faculty of Engineering, Universiti Putra Malaysia, Serdang 43400, Selangor, Malaysia; rosnahs@upm.edu.my
4   Department of Electrical and Electronic, Faculty of Engineering, Universiti Putra Malaysia, Serdang 43400, Selangor, Malaysia; hhizam@upm.edu.my
5   Department of Statistics, School of Mathematics and Statistics, University of New South Wales, Sydney 2052, Australia
6   Department of Crop Science, Faculty of Agriculture, Universiti Putra Malaysia, Serdang 43400, Selangor, Malaysia; jujunakasha@upm.edu.my
*   Correspondence: gs51010@student.upm.edu.my (N.F.O.); m_effendy@upm.edu.my (M.E.Y.)

**Abstract:** The increasing concerns about the impact of large-scale solar photovoltaic farms on the environment and the energy crisis have raised many questions. This issue is mainly addressed by the integration of agriculture advancement in solar photovoltaic systems infrastructure facilities, commonly known as agrivoltaic. Through the use of these systems, the production of crops can be increased, and the efficiency of PV panels can be improved. Accordingly, adopting such synergistic paths forward can contribute toward building resilient energy-generation and food-production systems. The utilization of cooling techniques can provide a potential solution for the excessive heating of PV cells and lower cell temperatures. Effective cooling applied to PV cells significantly improves their electrical efficiency, as well as increasing their lifespan because of decreasing thermal stresses. This paper shares an overview of both active and passive cooling approaches in solar PV applications with an emphasis on newly developed agrivoltaic natural cooling systems. Actual data analysis at the 2 MWp Puchong agrivoltaic farm shows a significant value of 3% increase of the DC generation (on average) which is most beneficial to solar farm operators.

**Keywords:** agriculture advancement; large scale solar; natural cooling; agrivoltaic; sustainability

## 1. Introduction

Across the globe, the amount of electricity produced by the large-scale solar (LSS) photovoltaic (PV) installations has shown an exponential growth in recent decades as concern has grown toward clean renewable energy for mitigating the energy crisis and environmental issues [1,2]. As an example, many LSS PV farms have been operated in Malaysia. The Malaysian government expects to achieve 45% deduction of $CO_2$, mainly by LSS PV farms, by 2030. Meanwhile, 10% of national electricity demands will be also satisfied through this continuous effort [3]. However, Barron-Gafford et al. [1] illustrated that LSS PV installations would cause a "heat island" effect. In other words, local surrounding temperatures over the LSS PV plant would increase. In some cases, therefore, the PV "heat island" effect has sparked public concerns and has indirectly led to resistance to the development of LSS PV farms.

As is known, PV cells generate electricity as well as heat. Up to 80% of the incident solar radiation can be absorbed by PV cells [4,5]. However, only a small portion of the

absorbed incident solar energy is converted into electricity via PV photonic effect. Much of the remaining energy is dissipated as heat above ambient temperature, depending upon the conversion efficiencies of the PV cell technology utilized as shown in Figure 1. The elevated temperatures can be caused by heat accumulating on the surface of the PV cells [5]. Accordingly, the operating temperature of the PV cells also linearly increases, resulting in one of the most important factors that can influence the PV cells' performance: irreversible degradation and shortening of the cells' lifetime [6,7]. It is clear that the use of cooling techniques on the PV is of great importance. Currently, numerous cooling technologies for regulating the thermal issue of PV systems have been investigated in many studies [5].

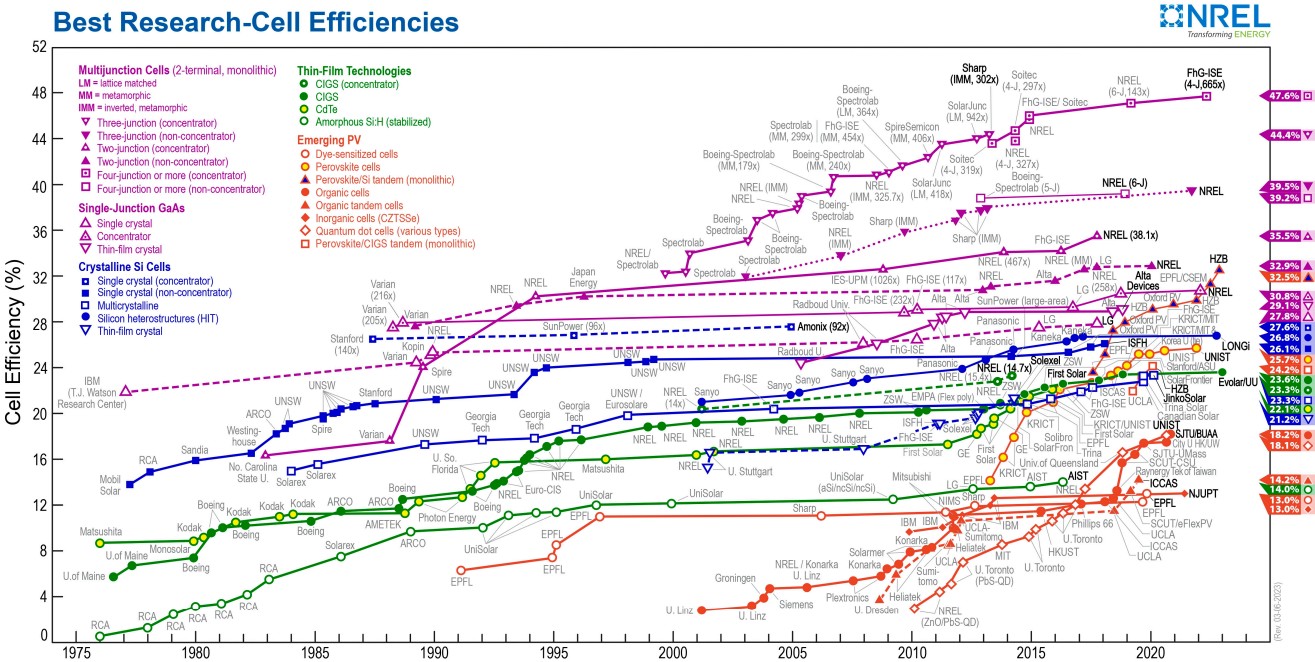

**Figure 1.** The cell efficiencies of various types of solar PV cells [8].

### 1.1. Six Common Cooling Approaches for Solar PV Cells

References [9,10] suggest that the temperature of the PV cells without cooling can increase up to 80 °C under warm and hot climate conditions. Depending on the PV cell technology used, every 1 °C increase in temperature of a PV module results in 0.4–0.5% reduction on the output power [9,11]. The utilization of cooling techniques can provide a potential solution for the excessive heating of PV cells and for lowering cell temperatures. Therefore, PV systems not only consist of inverters, as well as other electrical and mechanical devices, but also solar cell cooling [2,12]. Effective cooling of PV cells significantly improves their electrical efficiency, as well as increasing the lifespan of the PV cells because of the decreased thermal stresses. Approaches to cooling PV cells can be mainly classified as active and passive. Typically, the type of cooling (active or passive) approach and the materials adopted in cooling are selected in accordance with local weather conditions [13,14]. Figure 2 shows some active and passive cooling methods for PV cells.

#### 1.1.1. Active Cooling

Active cooling is a process that removes the heat from the system by using external coolant devices such as pump water, forced air, or fans to cool the panels. One of the drawbacks of active cooling is that a part of generated electrical energy is used by the external coolant system. However, the total output of the PV system with active cooling is higher than that with passive cooling and is also more effective for cooling heat transfer rates [14]. Some studies regarding active cooling methods, such as air cooling, water cooling, and Thermoelectric cooling for PV cells are shown in Table 1.

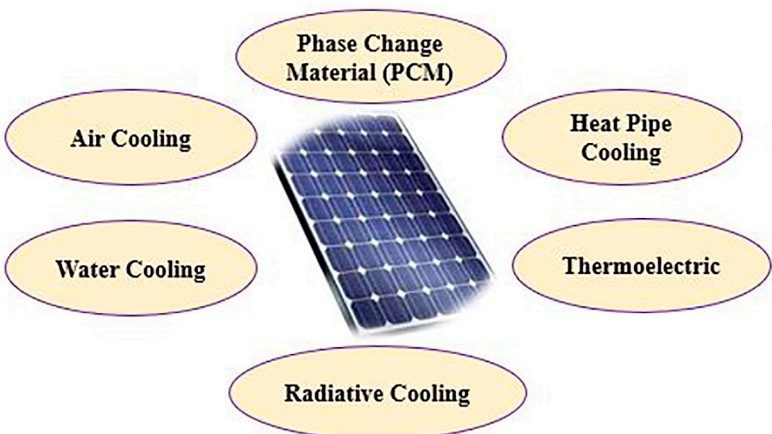

**Figure 2.** Six common cooling approaches for PV cells.

Air is regarded as a common cooling medium where air cooling systems are typically used in various devices for reducing the temperature and thermal management. Air cooling of PV cells is performed by using fans or other means to create forced convection airflow and then to decrease the temperature of PV cells. Despite the fact that using air as a coolant is less efficient than using liquids, air cooling offers some benefits, such as minimum material usage and cheap operating costs [15–17].

Water is the most frequently employed fluid in liquid-based cooling of PV systems for PV cells. The process of water cooling is performed by either spraying the water on the surface of the PV modules directly or passing the water behind the panel from inside the installed pipe [14]. Thermoelectric (TE) cooling technology is used to capture and convert excess heat from the PV cells directly into electricity. TE modules offer outstanding features of being lightweight, maintenance free, strongly reliable, noiseless in operation, and no complex parts. Thermoelectric generators (TEG) are one type of TE module that generates electrical power from the temperature gradient [5].

**Table 1.** A review of some studies on the active cooling methods for PV cells.

| Air Cooling Approach | | |
| --- | --- | --- |
| **Authors** | **Study Aim** | **Result** |
| Teo et al. [18] | To compare the performance of the PV module with and without air cooling. | With air cooling, the operating temperature of PV module could be kept at 38 °C and electrical efficiency maintained around 12.5%, whereas these two values can rise to around 68 °C and 8.6%, respectively, without air cooling. |
| Sajjad et al. [19] | To improve the performance efficiency of PV modules by using forced air cooling. | Compared to PV modules without cooling, forced air cooling achieves 6% and 7.2% power ratio and higher electrical efficiency, respectively. |
| **Water cooling approach** | | |
| **Authors** | **Study Aim** | **Result** |
| Krauter [20] | To investigate the impact of utilizing water flow as a coolant on the performance of the PV cells. | The temperature of cells was reduced to 22 °C by the water cooling approach, thus the output of the cells was improved by 10.3%. |
| Mah et al. [21] | To improve the performance of a crystalline silicon PV system via water cooling in a tropical region. | The power output of crystalline silicon PV system was increased by 15%, and each panel produced 0.0178 kW·h with the water cooling under the 1150 solar irradiance. In addition, water cooling also contributed to the uniform temperature distribution between the front and the back surfaces of the panels. |

**Table 1.** *Cont.*

| Thermoelectric cooling approach | | |
|---|---|---|
| **Authors** | **Study Aim** | **Result** |
| Sark [22] | To determine the efficiency of PV panel with TE converter by using a numerical model. | TE converter resulted in 8–23% enhancement of PV module's electrical efficiency. |
| Benghanem et al. [23] | To present the performance of PV cells by using TE module as the cooling system. | The temperature of PV cells dropped from 83 °C to 65 °C with TE modules, while the efficiency of PV cells decreased by 0.5% along with per °C rise in temperature. |

1.1.2. Passive Cooling

Passive cooling is a natural approach that provides air or liquid circulation to reduce the heat of the system, which is highly suitable for PV cooling projects. Unlike the aforementioned active cooling, passive cooling does not need to use any external power source for driving the cooling system, thus this benefit contributes to a simpler structure and lowers maintenance costs [24]. Some studies of passive cooling methods such as phase change material (PCM), heat pipe cooling, and radiative cooling are demonstrated in Table 2.

PCM is a useful passive cooling approach in the thermal management of PV cells due to its great capacity for heat storage with prolonged heat availability. PCM absorbs extra heat from PV cells through its latent heat, then keeps PV cells at the accepted temperature for a certain period. The melting temperature of PCM for thermal management of PV cells is recommended to be 25 °C in the summer. Additionally, the PCM's melting temperature should be lower than the PV cells' temperature for effective thermal management of the cells [25]. Heat pipes as coolant devices for PV cells are used due to their high thermal conductivity, uncertain heat flux, and ability to create uniform temperatures. Such devices are typically composed of a sealed pipe with high thermal conductivity material at both condenser and evaporator. Heat pipes can lower the temperature and then enhance the electrical efficiency of PV cells by transferring heat from PV cells to water or air [12,14]. Radiative cooling is a passive cooling method based on using an atmospheric window with a transparency in the wavelength range between 8 μm and 14 μm. In other words, radiative cooling is only achievable when the entrance heat flux caused by conduction, convection or radiation to the infrared spectral layer (with the thickness between 8 μm and 14 μm) is smaller than the output heat flux from the earth's body. It is valuable to note that the spectral alteration of the emissivity of modules' areas for thermal radiation and absorption determines the rate of radiative cooling [14,26].

**Table 2.** A review of some studies on the passive cooling methods for PV cells.

| PCM Approach | | |
|---|---|---|
| **Authors** | **Study Aim** | **Result** |
| Hasan et al. [27] | To study the performance of cooling the PV cell by using the paraffin-based PCM with melting temperature (38 °C–43 °C). | PCM cooling dropped 10.5 °C in PV temperature on average at peak time and contributed to increasing 5.9% in PV output power on annual basis. |
| Wongwuttanasatian et al. [28] | To investigate the performance of PV system by using palm wax as a low-cost PCM. | The temperature of PV system was decreased about 6.1 °C along with a 5.3% electrical efficiency increase via PCM cooling, which compared to the PV system without the cooling. |

**Table 2.** *Cont.*

| Heat pipe cooling approach | | |
|---|---|---|
| **Authors** | **Study Aim** | **Result** |
| Habeeb et al. [29] | To carry out the performance of cooling PV panels using thermosyphon heat pipe at Baghdad climate. | Compared to the traditional panel, the module temperature was colder at a rate of 15–35%, and its efficiency was enhanced by 11–14%. Under 1000 $Wm^{-2}$ solar irradiation, the generated electricity was increased to 18% with the heat pipe cooling. |
| Alizadeh et al. [30] | To investigate the thermal performance for PV cooling by using pulsating heat pipe. | |
| Radiative cooling approach | | |
| **Authors** | **Study Aim** | **Result** |
| Nishioka et al. [31] | To use a high radiative coating to improve the performance of CPVs. | The temperature of the solar cells reduced around 10 °C, and their efficiency increased about 0.5%. In addition, the uniform temperature distribution in the cells was improved. |
| Zhu et al. [32] | To utilize a radiative cooling strategy that consists of a sky-access photonic thermal emitter to avoid the high operating temperature of PV cells without affecting their absorption coefficient. | The temperature of this design was forecast to drop at 17.6 °C and thus enhanced the electrical efficiency up to 7.9% under 800 $Wm^{-2}$ solar intensity. |

Even though the aforementioned cooling methods have remarkable effects on improving the performance of PV cells in terms of temperature reduction and electrical efficiency, the overall investment cost of cooling systems is a considerable concern when dealing with LSS PV farms [26]. Additionally, employing gravel as ground cover for PV installations is a business-as-usual approach. The ground-mounted PV installations with gravel ground cover also create a "heat island" effect. In other words, temperatures around PV solar arrays increase. Replacing the gravel with vegetation by strategic planning, therefore, can help to counter the heat feedback loop. As such, agrivoltaic technology holds promising implications for the food-energy-water nexus [33].

*1.2. Agrivoltaic Approach*

The co-location of PV and agriculture, commonly known as an agrivoltaic system, offers a win-win solution through many benefits, such as reducing water loss, increasing crop production, and improving the conversion efficiency of PV panels. Accordingly, adopting such synergistic paths forward can contribute toward building resilient energy-generation and food-production systems [33].

In the USA, Barron-Gafford et al. [34] performed an agrivoltaic system by planting chiltepin peppers, jalapeños, and cherry tomatoes under PV arrays. The system was created to capture the effects of this approach on physical and biological features during the average three-month summer growing season. Compared to the traditional planting area (control) of the food production, the total productions of chiltepin peppers and cherry tomatoes in the agrivoltaic system were three and two times greater, respectively. With regard to water savings, as shown in Figure 3(i), soil moisture remained around 15% and 5% higher for irrigating every two days and irrigating every day before the next watering, respectively, in the agrivoltaic system. With regard to the improved renewable energy production, as shown in Figure 3(ii), PV panels in the agrivoltaic system were approximately 9 °C cooler during daytime hours. On balance, therefore, the agrivoltaic approach provides mutual benefits in drylands in terms of the food-energy-water nexus.

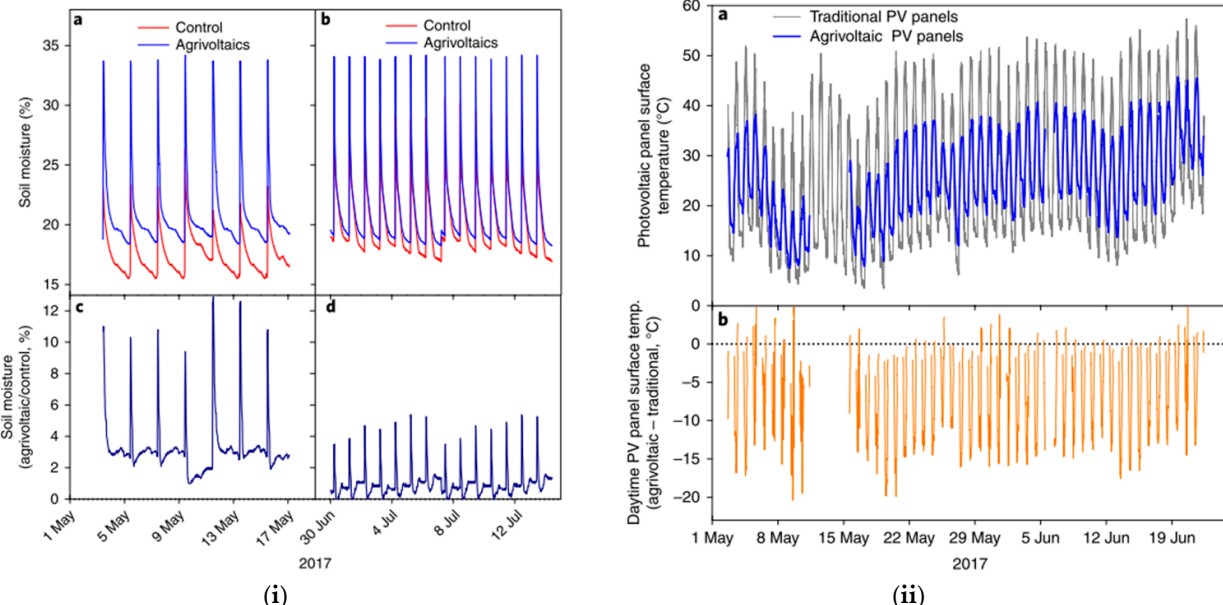

**Figure 3.** (**i**) Impacts of agrivoltaic over control installations on soil moisture: (**a**,**b**)—the comparison between control plots and agrivoltaics on soil moisture; (**c**,**d**)—differences between soil moisture in the agrivoltaic and in the control settings where positive values mean additional moisture in the agrivoltaic setting. (**ii**) impacts of agrivoltaic over traditional ground-mounted installations on the surface temperature of PV panels: (**a**)—the comparison between the surface temperature of traditional PV panels and the surface temperature of agrivoltaic PV panels; (**b**)—differences in PV panel temperature between the agrivoltaic and traditional settings where negative values mean the degree to which PV panels in the agrivoltaic were cooler. Reproduced with permission [34]. Copyright 2018, Springer Nature.

In Europe, agriculture technology company Sun'Agri from France showed that the trees shaded by the agrivoltaic system in the Durance Valley decreased the ambient temperature from 2 °C to 4 °C and also contributed to 63% reduction of water stress on the crops [35]. In China, the capacity of a 640 MW solar park was installed, while goji berries were planted under the solar panels. The results showed that the evaporation of land moisture for this solar park effectively reduced by 30–40%, and 85% vegetation coverage significantly improved the climate in this region. More interestingly, the ecosystem has also changed accordingly in this region. For instance, the number of small wild animals, such as hares, pheasants, and sparrows, has significantly increased [36]. In Singapore, Teng et al. [37] investigated the impact of agrivoltaic system on the surrounding rooftop microclimate by using ENVI-met simulation. Compared to the results without crops, on sunny days under the agrivoltaic approach, PV temperatures were on average reduced by 2.83 °C, and PV efficiency performance was improved by 1.13–1.42%. On cloudy days under the agrivoltaic approach, PV temperatures were also on average lower by 0.71 °C, and PV efficiency performance was enhanced by 0.28–0.35%.

An illustration of the practical implementation of agrivoltaic projects is shown in Figure 4 with the geographical details distributed worldwide. Herein, this paper aims to perform the tropical field validation for energy performance via agrivoltaic natural cooling approach in the LSS PV farm in Malaysia. The structure of this paper is as follows: Section 2 contains the details of field setup, data logging, and experimental approach. Section 3 shares some field analysis on the environmental parameters, namely ambient temperature, wind speed, and relative humidity for both weather stations. The main findings on herbal natural cooling in large scale solar PV farms is described based on Fisher ANOVA on energy at different plots and with some statistical justifications to support the results. Section 4 concludes the study with a significant DC energy increase via agrivoltaic

approach as per the Welch two sample *t*-test between energy production at agrivoltaic and non-agrivoltaic plots.

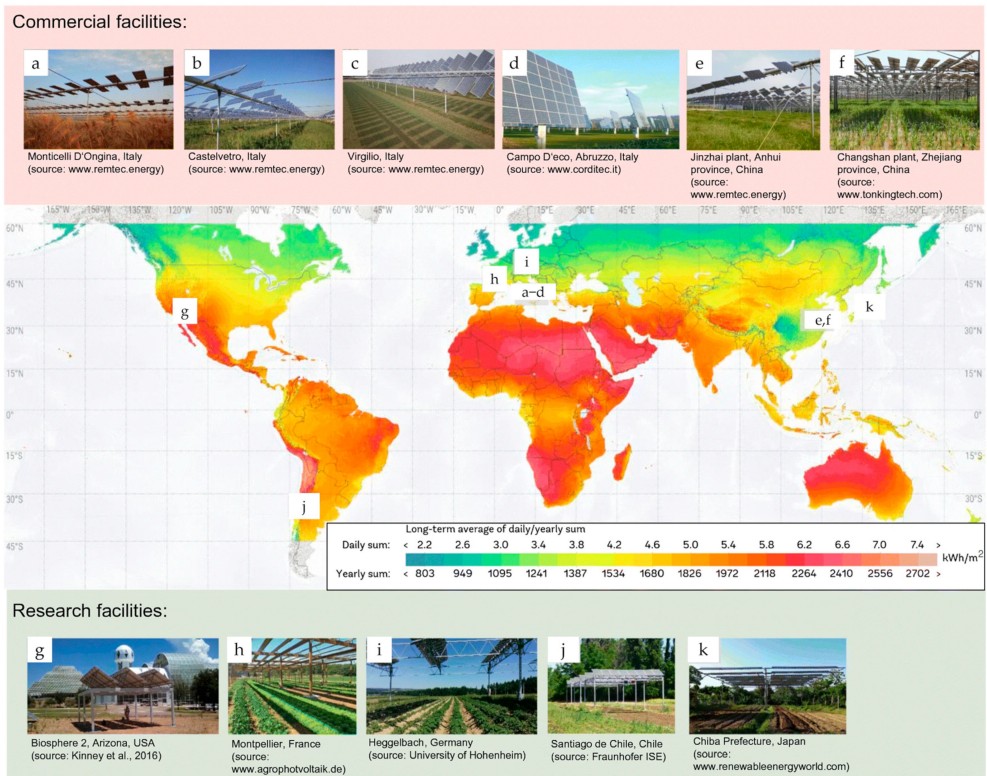

**Figure 4.** Geographical distribution of agrivoltaic projects for industrial and research facilities worldwide [38]. a; Monticelli D'Ongina Italy (www.remtec.energy (accessed on 22 March 2023)), b; Castelvelo Italy (www.remtec.energy (accessed on 22 March 2023)), c; Virgilio Italy (www.remtec.energy (accessed on 22 March 2023)), d; Campo D'eco Abruzzo Italy (www.corditec.it (accessed on 22 March 2023)), e; Jinzhai plant Anhui Province China (www.remtec.energy (accessed on 22 March 2023)), f; Changshan plant Zhejiang Province China (www.tonkingtech.com (accessed on 22 March 2023)), g; Biosphere 2, Arizona, USA [39], h; Montpellier France (www.agrophotvoltaik.de (accessed on 22 March 2023)), k; Chiba Prefecture Japan (www.renewableenergyworld.com (accessed on 22 March 2023)). Copyright 2019, Springer.

## 2. Methodology

### 2.1. Site Setup

The LSS PV field setup was located at UPM Agri Solar Power Plant in Puchong, Selangor, with 2 MWp generating capacity. It consisted of 8064 monocrystalline PV modules within five acres area, including 84 strings segregated into 12 plots. Each PV plot was divided by seven strings and could be further separated into four sections. As shown in Figures 5 and 6, plots were selected for the analysis of this research. The plots (rectangles in round yellow dots) were planted with Misai Kucing. Plot 7 is designated as the reference plot, and the condition is maintained as per normal Solarfarm structures without any Misai Kucing crops planted underneath. Figure 6 presents the experimental facilities of this research. Figure 7 shows the data logging platform via Sunny Explorer software from SMA Solar Technology AG for electrical output.

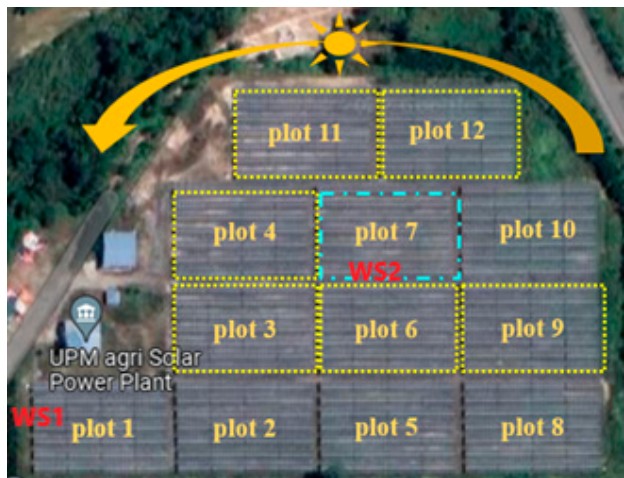

**Figure 5.** Google map layout of LSS PV farm in Puchong, Selangor (five acres). WS is the location for Weather Station.

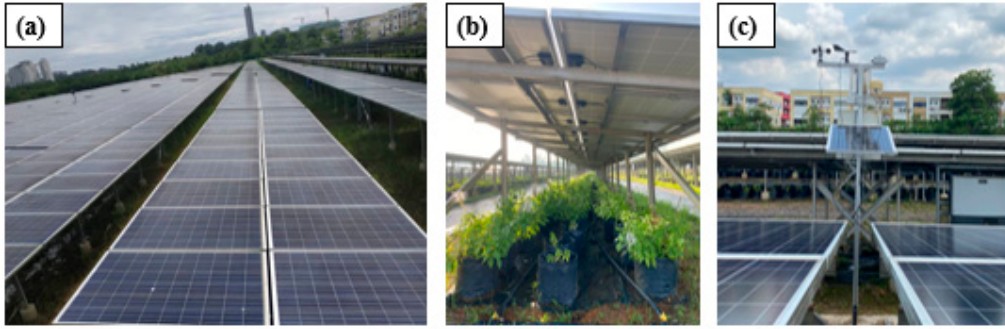

**Figure 6.** (**a**) LSS PV farm setup facing south with 5° slanting angle; (**b**) Misai Kucing plants were planted beneath the PV panels as agrivoltaic setup; (**c**) the weather station for LSS PV farm.

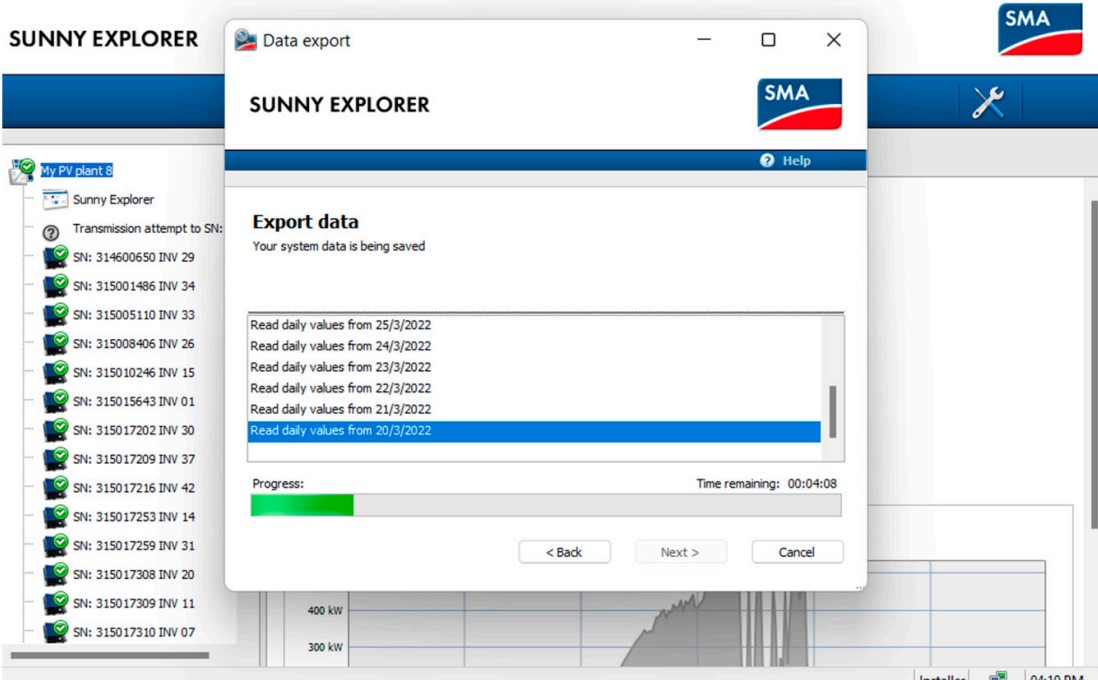

**Figure 7.** Data logging platform using Sunny Explorer for electrical output.

### 2.2. Experiment Process

The experiment was conducted from February 2022 to March 2022 for two months with continuous monitoring of all plots. The data logging process occurred from 7:00 a.m. to 7:00 p.m. each day (only during the presence of sunlight). Figure 8 shows PV modules' construction at Puchong Solarfarm with mounting structures.

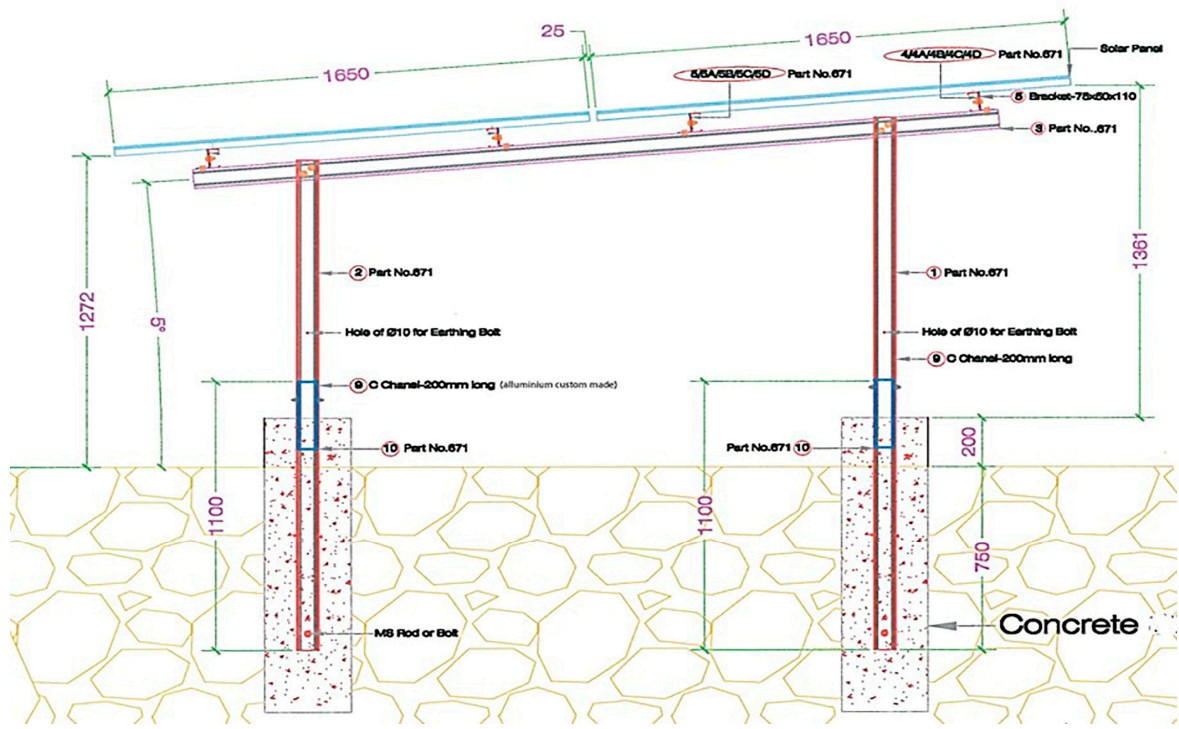

**Figure 8.** PV modules mounting structure with height distance from ground level.

Some descriptive statistics to obtain a preliminary understanding of energy production under different settings (e.g., sun level and plot) were computed. To assess the relationship between sun levels and energy production, we conducted a Welch one-way analysis of variance (ANOVA) [40] and a further pairwise comparison (via Games-Howell test) [41]. Next, to assess the relationship between agrivoltaic status and energy production, we conducted a Fisher ANOVA [42] and further pairwise comparison with reference to the non-agrivoltaic plot (via Student's *t* pairwise comparison test with Dunnett's method adjustment) [43]. Finally, to further look into the comparison between agrivoltaic and non-agrivoltaic plots, we grouped all agrivoltaic plots into a single group and conducted a two-sample mean test.

## 3. Results and Discussion

Based on the collected energy data for Puchong Solarfarm, it is observed that only plots 3, 4, 6, 7, 9, 11 and 12 (as shown in Figure 5) have recorded continuous data without errors. Thus, those without faulty data were used for analysis. These plots are considered sufficient based on their location surrounding the reference plot. The typical results on environmental parameters are further analyzed based on the two-sample location, i.e., near Plot 1 (agrivoltaic:WS1-corner) and near Plot 7 (non-agrivoltaic: WS2-middle). Based on the 24 h data collection, the ambient temperature in the agrivoltaic area and at the middle of the farm area is nearly the same with the maximum value of 41.2 °C recorded at the middle area as shown in Figure 9. The wind profiling for both locations is almost the same throughout the sample day as shown in Figure 10. As for the relative humidity shown in Figure 11, the value recorded for agrivoltaic areas shows a much higher humidity level with an average difference of 5%.

Higher humidity near solar panels can affect the lifespan of solar panels in several ways. First, increased humidity can lead to corrosion of metal parts and electrical components in the panels. Second, high humidity can cause moisture to enter the solar cells, which can degrade their performance and efficiency over time. Additionally, high humidity can promote the growth of microorganisms, such as algae, which can accumulate on the surface of solar panels and reduce their efficiency. Finally, high humidity can also lead to the formation of condensation on the surface of the panels, which can damage the electrical components and reduce the efficiency of the system. To mitigate the effects of high humidity on solar panels, it is important to ensure that the panels are properly sealed and protected from moisture, which is a critical parameter during the PV manufacturing process [44–47]. In this study, the focus will be on temperature reduction via the natural cooling approach. Regular cleaning of the panels can also help to prevent the accumulation of algae and other debris on the surface of the panels, which can reduce their efficiency over time.

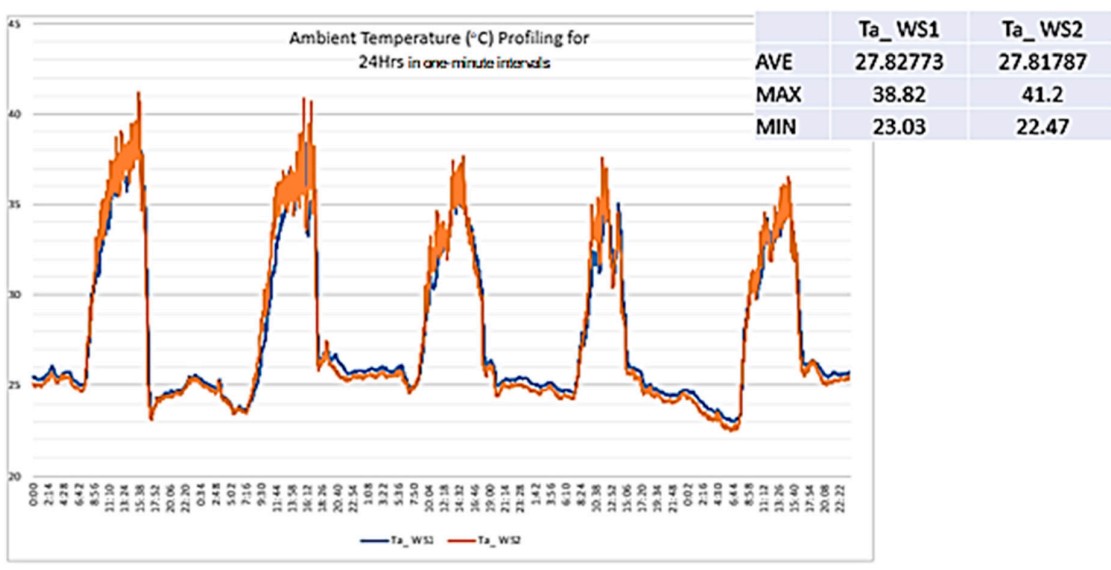

**Figure 9.** Ambient temperature profiling.

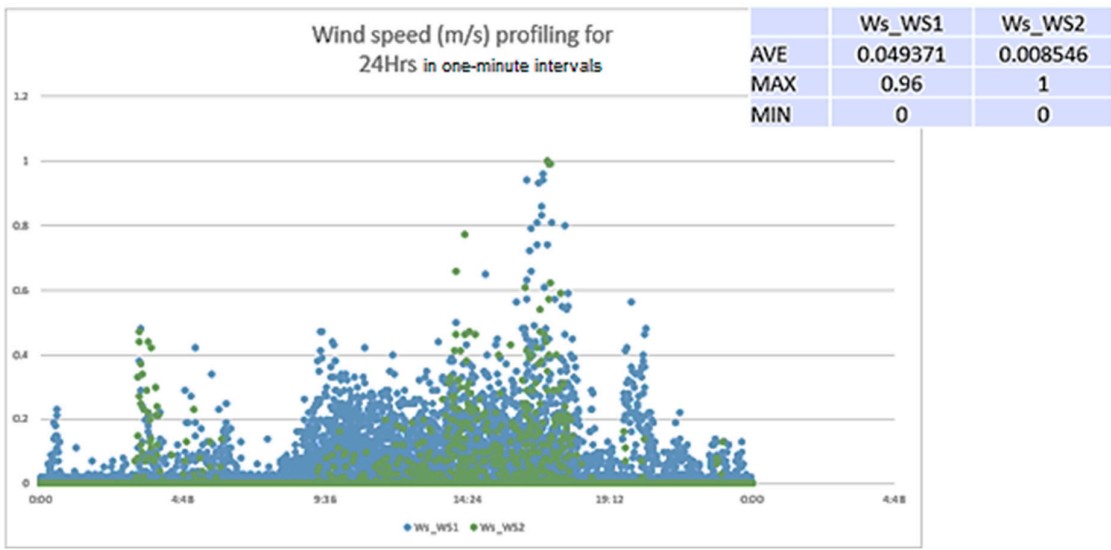

**Figure 10.** Wind speed profiling.

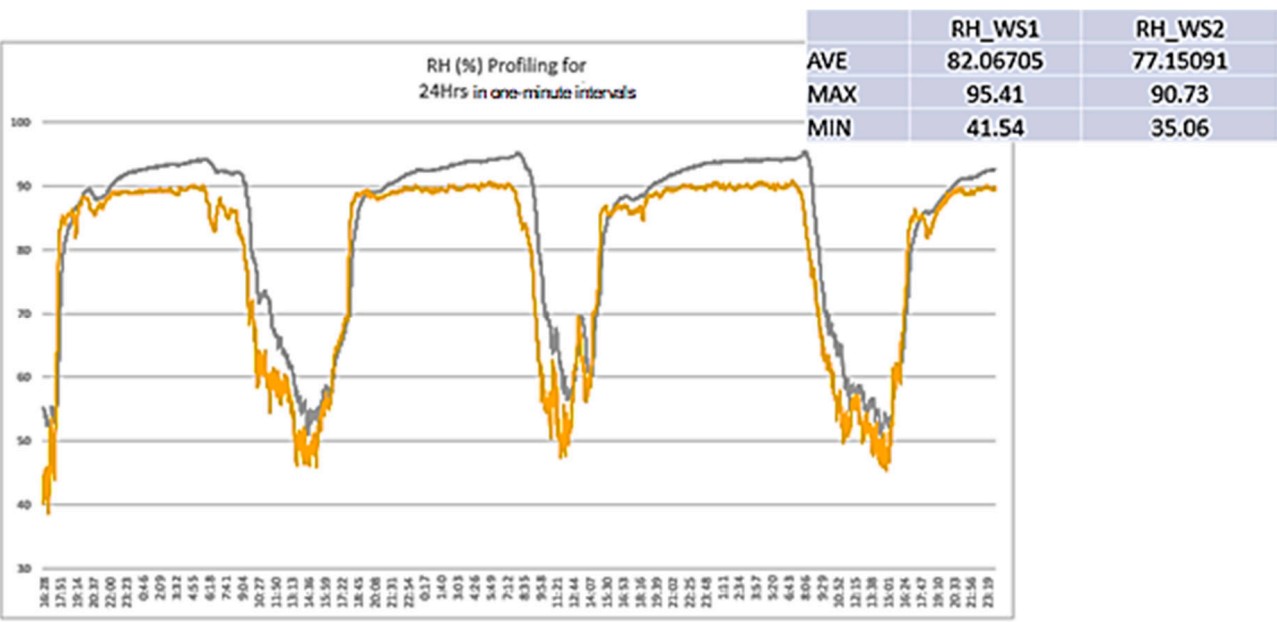

**Figure 11.** Relative humidity profiling.

A preliminary observation from descriptive statistics showed that Plot 7 (the non-agrivoltaic plot) had the lowest energy, and most energy was captured during peak sun (refer to Appendix A Table A1 for further detailed descriptive statistics). Some further analyses were conducted to examine the different energy level captured at different sun levels of the day as shown in Appendix A Figure A1. First, several main assumptions about the distribution of energy at different sun levels that were tested were the assumption of normality as well as constant variance of residuals (refer to Appendix B Figure A2). Since we have a large dataset, normality is assumed. Unfortunately, there is a fan-shape in the residuals vs. fitted plot which shows inconsistency in variance (heteroscedasticity) between sun levels. We further proved the presence of heteroscedasticity through the Bartlett test [48] of homogeneity of variances (Bartlett's K-squared = 8275.1, df = 4, $p$-value < $2.2 \times 10^{-16}$). Hence, Welch ANOVA was employed (refer to Appendix B Figure A3). There is a significant difference ($F_{\text{Welch}}$ = 26,408.92; $p$-value = 0.00 < 0.05) in energy at different sun levels. The Games-Howell test showed that the energy is different between pairs of sun levels (all $p$-values < 0.05) (refer to Appendix B Table A2). That is, the energy level is different at different sun levels, and the highest was recorded during the peak sun, like that obtained in Othman et al. [49]. The sample comparative analysis of both agrivoltaic and non-agrivoltaic plots is shown in Figure 12 for two sample plots and Table 3 with details of the electrical outputs for 7 sample plots, which confirms the increase in DC electricity generation by means of natural plant cooling. Based on actual field data analysis, the location of each plot showed some varying values although they are installed at the same location. As an example, Plot 3 generated a higher power than Plot 9, with different fluctuation patterns throughout the day along with the sun movement where the location of PV module distribution does provide some impact in terms of DC generation. Shadow is not a factor because the location was selected and constructed to negate any shadow impact. With respect to Plot 7 as the non-agrivoltaic plot, this study has proven a significant increase in DC power generation.

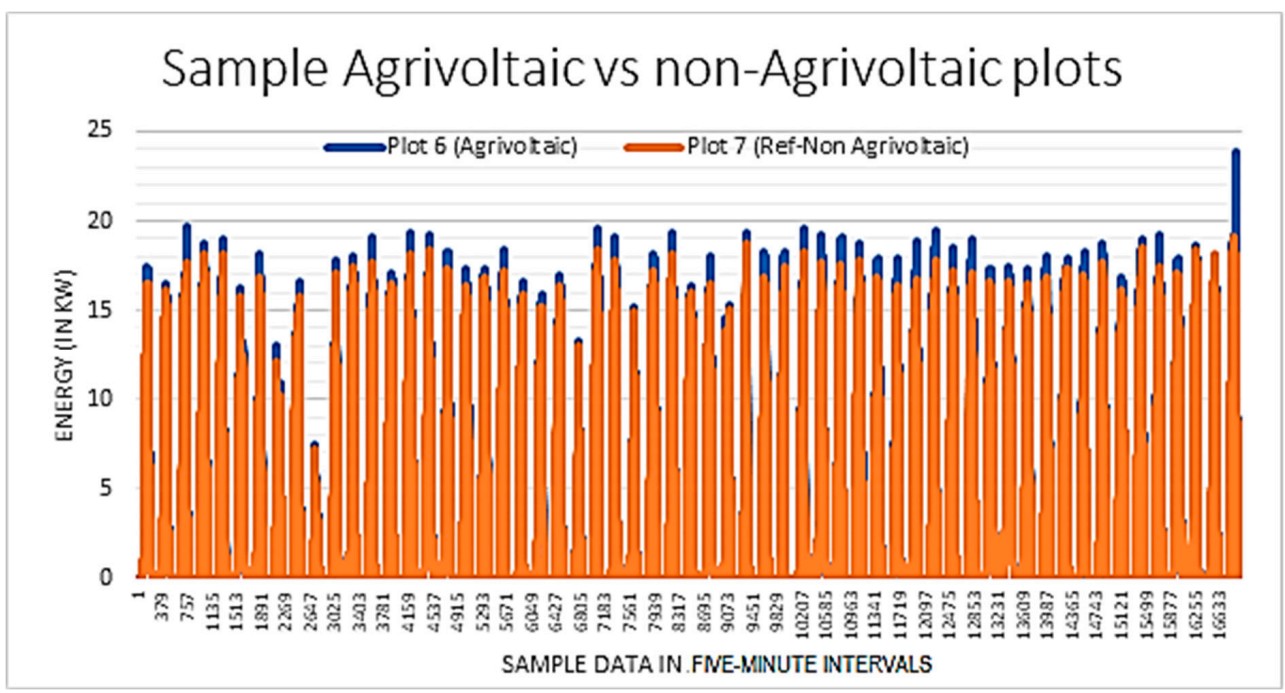

**Figure 12.** Sample agrivoltaic vs. non-agrivoltaic plots.

**Table 3.** The electrical output for the seven selected plots.

| | Plot 3 (Agrivoltaic) | Plot 4 (Agrivoltaic) | Plot 6 (Agrivoltaic) | Plot 7 (Ref-Non Agrivoltaic) | Plot 9 (Agrivoltaic) | Plot 11 (Agrivoltaic) | Plot 12 (Agrivoltaic) |
|---|---|---|---|---|---|---|---|
| Sum (kW) | 66,031.087 | 66,176.308 | 64,724.269 | 63,835.959 | 64,652.976 | 65,161.215 | 64,907.0955 |
| Ave (kW) | 3.886239009 | 3.894785946 | 3.809326644 | 3.757045436 | 3.805130716 | 3.835042964 | 3.82008684 |
| Max (kW) | 19.758 | 19.632 | 23.808 | 19.104 | 18.396 | 19.278 | 18.7275 |
| Comparison | P3-P7 | P4-P7 | P6-P7 | P9-P7 | P11-P7 | P12-P7 | |
| kW | 2195.128 | 2340.349 | 888.31 | 817.017 | 1325.256 | 1071.1365 | |
| % | 3.324385679 | 3.536536067 | 1.372452735 | 1.263695889 | 2.033811064 | 1.650261026 | |

The analysis of the effect of agrivoltaic plots on energy production begins with some assumptions required for the parametric statistical tests: i.e., normality and constant variance of residuals were fulfilled. Hence, Fisher ANOVA was used, and some reliable results are summarized in Figure 13.

There is strong evidence for the difference ($F_{Fisher}$ = 2.39; *p*-value = 0.03 < 0.05) in energy from different plots. Student's *t* pairwise comparison test with Dunnett's method adjustment (a close approximation to the Dunnett adjustment) showed that the energy is different between Plot 7 and two other plots, i.e., Plot 3 and Plot 4, at 95% confidence level (refer to Table 4). The 95% confidence intervals of the mean difference between individual agrivoltaic plots (Plots 3, 4, 9 and 11) and non-agrivoltaic plot (Plot 7) further support the results, i.e., the intervals for the pairs do not include zero (refer to Figure 14).

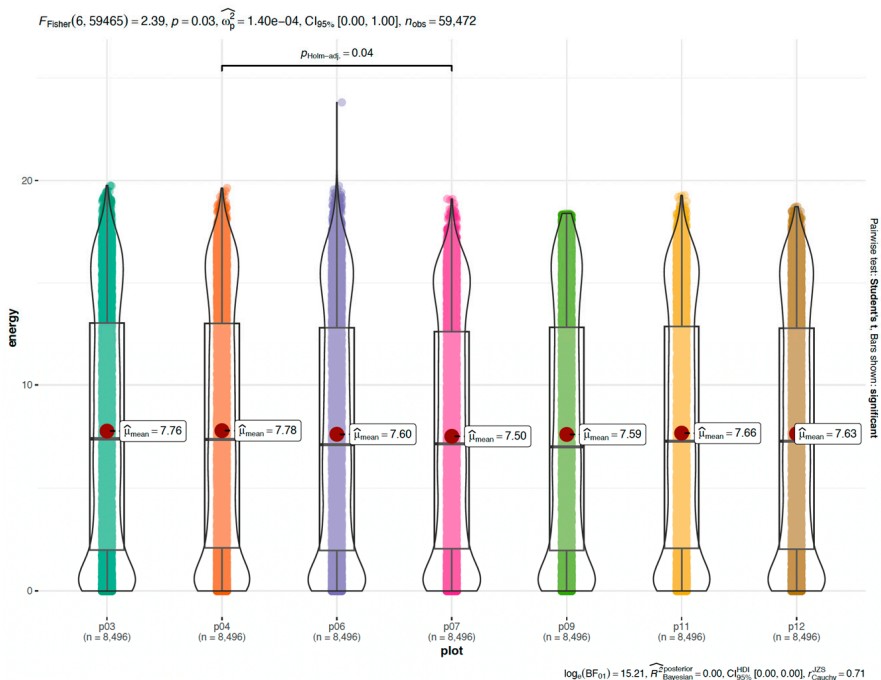

**Figure 13.** Fisher ANOVA on energy at different plots.

**Table 4.** Pairwise comparison test (with Dunnett's method adjustment) on energy from different plots (SE = 0.08792941; df = 59465).

| Contrast | Estimate | *t*-Ratio | Confidence Interval | *p*-Value |
|----------|----------|-----------|---------------------|-----------|
| p04–p07 | 0.2751 | 3.1292 | (−0.0826, 0.5986) | 0.0097 |
| p03–p07 | 0.2559 | 2.9105 | (−0.0591, 0.6221) | 0.0193 |
| p06–p07 | 0.1016 | 1.1552 | (0.0536, 0.7348) | 0.6825 |
| p09–p07 | 0.0931 | 1.0591 | (−0.0051, 0.6761) | 0.7406 |
| p11–p07 | 0.1563 | 1.7779 | (−0.0476, 0.6336) | 0.2992 |
| p12–p07 | 0.1247 | 1.4185 | (−0.6604, 0.0209) | 0.5128 |

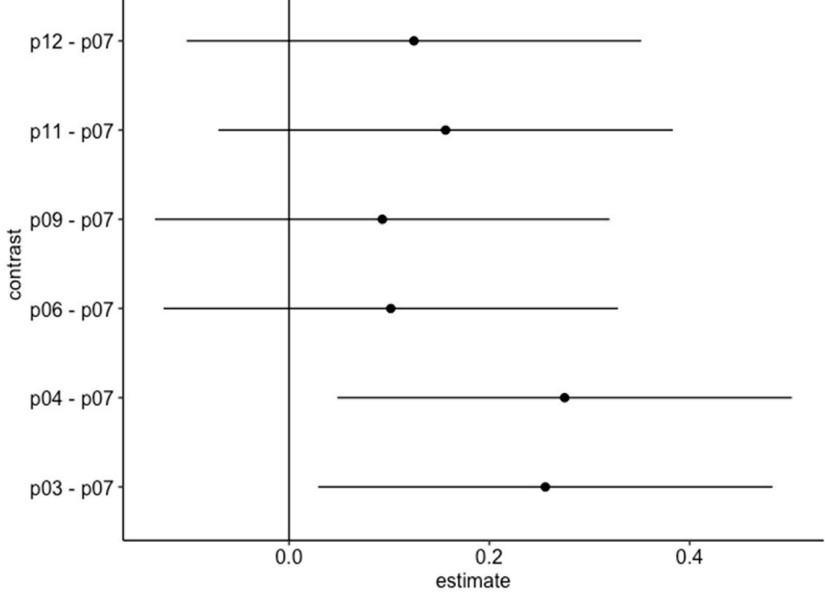

**Figure 14.** 95% confidence interval of mean difference between Agrivoltaic plots and non-Agrivoltaic plot.

Further, the two-sample mean comparison (Welch t = 2.5624, df = 11,743, *p*-value = 0.01041 < 0.05, 95% confidence interval = [0.0394, 0.2962]) shows strong evidence for the difference between a group of agrivoltaic plots and a non-agrivoltaic plot (refer Table 5). In summary, the highest average energy increase by plot and the overall plot energy average increase are 0.28 kW (3.73%) and 0.17 kW (2.24%), respectively, with the implementation of agrivoltaic.

**Table 5.** Welch two sample *t*-test between energy production at agrivoltaic and non-agrivoltaic plots.

| *t* | df | Confidence Interval | *p*-Value |
|---|---|---|---|
| 2.5624 | 11743 | [0.0394, 0.2962] | 0.0104 |

## 4. Conclusions

The agrivoltaic integration in large-scale solar PV farms adopts such synergistic paths forward which contribute toward building resilient energy-generation and food-production systems. This study provides the proof of concept where herbal plantation (in this case, Misai Kucing cultivation) as a means of agrivoltaic plot supports the operation of solar farms through natural cooling directly underneath the solar PV arrays. The significant DC energy increase of 3% (on average) via agrivoltaic cooling provides sufficient savings and surplus to the operators plus some means of secondary business with the fresh produce. Thus, it is greatly recommended that all large-scale solar farms, especially in Malaysia, should be transformed, not only by producing electricity for the grid, but also by integrating agriculture via agrivoltaic approach. Future recommendations on the impacts of PV module lifespan, economic perspective, and soil properties under agrivoltaic conditions are suggested.

**Author Contributions:** Conceptualization, N.F.O., M.E.Y. and H.H.; methodology, L.L., A.S.M.S. and H.H.; software, A.H.J.; validation, M.E.Y. and A.H.J.; formal analysis, A.H.J.; investigation, L.L.; resources, N.F.O.; data curation, A.H.J. and J.N.J.; writing—original draft, N.F.O. and L.L.; writing—review & editing, M.E.Y.; visualization, A.S.M.S.; supervision, R.S. and J.N.J.; project administration, N.F.O. and R.S.; funding acquisition, M.E.Y. All authors have read and agreed to the published version of the manuscript.

**Funding:** The authors delegate our thanks to the Ministry of Energy and Natural Resources (MENR) under the AAIBE Research Fund (Vote no. 6300921) and the Research Management Centre (RMC), Universiti Putra Malaysia for the approval of research funding under the Putra Grants Scheme (Vote no. 9709000).

**Institutional Review Board Statement:** Not Applicable.

**Data Availability Statement:** Not Applicable.

**Conflicts of Interest:** The authors declare no conflict of interest.

## Appendix A. Descriptive Statistics

**Table A1.** Descriptive statistics of energy at different sun levels and plots.

| Plot | Measures | Sun Level | | | | | |
|---|---|---|---|---|---|---|---|
| | | Early Sun | Mild Sun (Evening) | Moderate Sun (Evening) | Moderate Sun (Morning) | Peak Sun | Overall |
| | | | | Plot 3 | | | |
| | Mean (SD) | 2 (2.41) | 2.75 (2.8) | 9.24 (5.13) | 10.55 (3.87) | 13.24 (4.07) | 7.76 (5.83) |
| | Interval [min, max] | [0, 10.89] | [0, 12.29] | [0, 19.74] | [0.72, 18.85] | [1.77, 19.76] | [0, 19.76] |
| | | | | Plot 4 | | | |
| | Mean (SD) | 2.22 (2.5) | 2.66 (2.75) | 9.1 (5.13) | 10.68 (3.84) | 13.2 (4.07) | 7.78 (5.8) |
| | Interval [min, max] | [0, 11.53] | [0, 12.12] | [0, 19.63] | [0.75, 18.74] | [1.82, 19.26] | [0, 19.63] |

**Table A1.** *Cont.*

| Plot | Measures | Sun Level | | | | | |
|---|---|---|---|---|---|---|---|
| | | Early Sun | Mild Sun (Evening) | Moderate Sun (Evening) | Moderate Sun (Morning) | Peak Sun | Overall |
| | | Plot 6 | | | | | |
| | Mean (SD) | 1.94 (2.32) | 2.75 (2.81) | 9.16 (5.16) | 10.24 (3.79) | 12.96 (4.13) | 7.6 (5.75) |
| | Interval [min, max] | [0, 10.84] | [0, 12.37] | [0, 23.81] | [0.7, 18.79] | [1.71, 19.42] | [0, 23.81] |
| | | Plot 7 (reference plot) | | | | | |
| | Mean (SD) | 2.14 (2.41) | 2.59 (2.69) | 8.8 (4.92) | 10.33 (3.68) | 12.67 (3.83) | 7.5 (5.56) |
| | Interval [min, max] | [0, 11.23] | [0, 11.71] | [0, 18.48] | [0.73, 18.41] | [1.76, 19.1] | [0, 19.1] |
| | | Plot 9 | | | | | |
| | Mean (SD) | 1.9 (2.27) | 2.73 (2.8) | 9.12 (5.17) | 10.27 (3.85) | 12.98 (4.22) | 7.59 (5.78) |
| | Interval [min, max] | [0, 10.78] | [0, 12.59] | [0, 18.34] | [0.72, 18.24] | [1.67, 18.4] | [0, 18.4] |
| | | Plot 11 | | | | | |
| | Mean (SD) | 2.2 (2.48) | 2.59 (2.7) | 8.94 (5.03) | 10.57 (3.78) | 12.97 (3.97) | 7.66 (5.7) |
| | Interval [min, max] | [0, 11.27] | [0, 11.84] | [0, 19.26] | [0.71, 18.6] | [1.79, 19.28] | [0, 19.28] |
| | | Plot 12 | | | | | |
| | Mean (SD) | 2.05 (2.37) | 2.66 (2.73) | 9.03 (5.01) | 10.42 (3.75) | 12.98 (3.92) | 7.63 (5.69) |
| | Interval [min, max] | [0, 10.74] | [0, 11.78] | [0, 18.72] | [0.72, 18.27] | [1.74, 18.73] | [0, 18.73] |
| | | Overall | | | | | |
| | Mean (SD) | 2.07 (2.4) | 2.67 (2.75) | 9.06 (5.08) | 10.44 (3.79) | 13 (4.04) | 7.65 (5.73) |
| | Interval [min, max] | [0, 11.53] | [0, 12.59] | [0, 23.81] | [0.7, 18.85] | [1.67, 19.76] | [0, 23.81] |

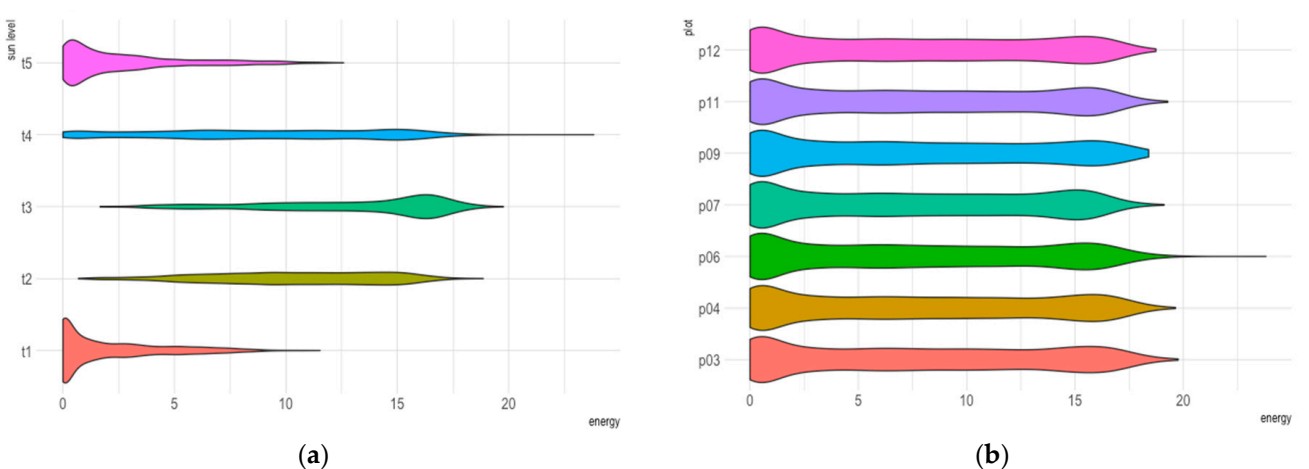

(**a**)                                     (**b**)

**Figure A1.** (**a**) Distribution of energy at different sun level. (**b**) Distribution of energy at different plots.

## Appendix B. Energy Level at Different Sun Levels

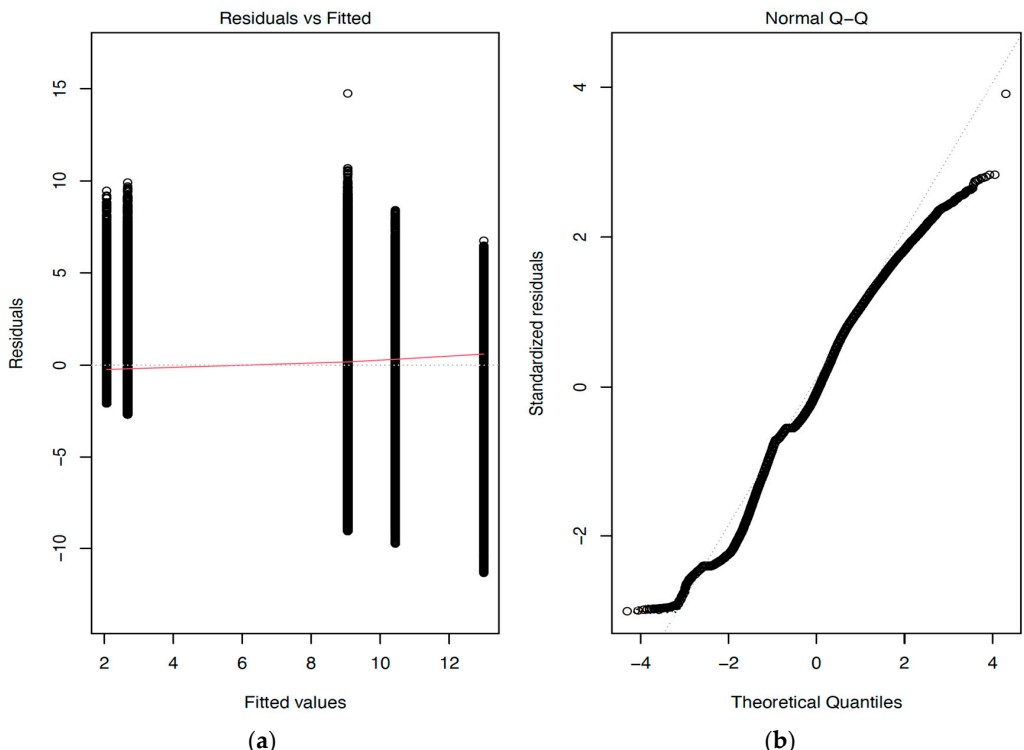

(**a**)

(**b**)

**Figure A2.** Equal variance and normality assessments on energy at different sun levels. (**a**) Residuals vs. fitted. (**b**) Normal Q-Q.

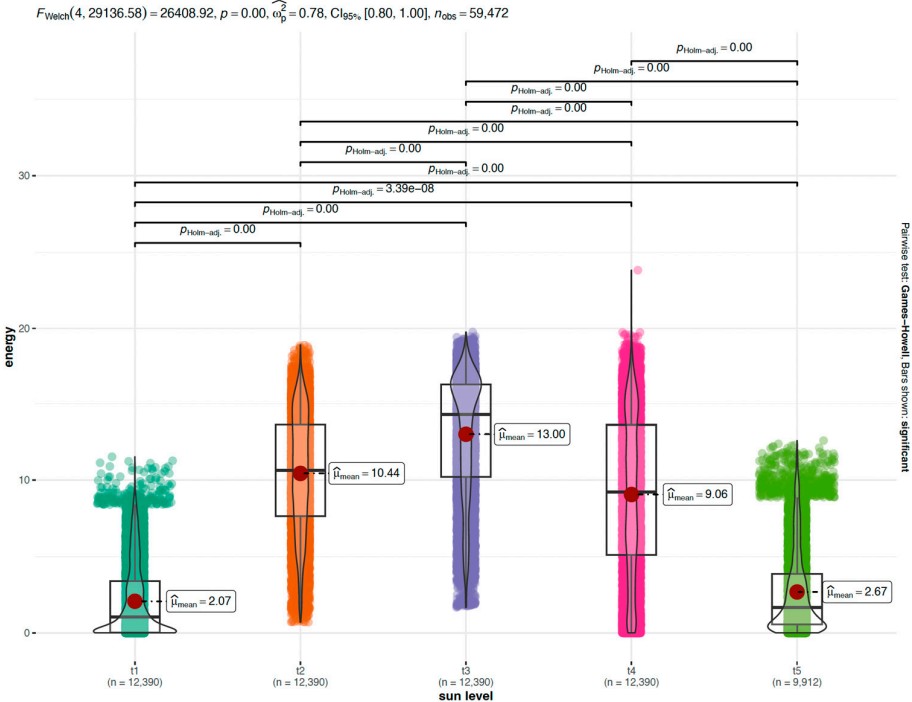

**Figure A3.** Welch ANOVA on energy at different sun levels.

**Table A2.** Pairwise comparison in energy at different sun levels using Games-Howell test.

| Contrast Pair | Estimate | Confidence Interval | *p*-Value Adjusted |
|---|---|---|---|
| t1–t2 | 8.3706 | [8.2607, 8.4806] | 0.0000 |
| t1–t3 | 10.9357 | [10.8207, 11.0508] | 0.0000 |
| t1–t4 | 6.9907 | [6.853, 7.1284] | 0.0000 |
| t1–t5 | 0.6094 | [0.5137, 0.705] | 0.0000 |
| t2–t3 | 2.5651 | [2.4293, 2.7008] | 0.0000 |
| t2–t4 | −1.3800 | [−1.5354, −1.2246] | 0.0000 |
| t2–t5 | −7.7613 | [−7.881, −7.6415] | 0.0000 |
| t3–t4 | −3.9450 | [−4.1041, −3.786] | 0.0000 |
| t3–t5 | −10.3263 | [−10.4507, −10.2019] | 0.0000 |
| t4–t5 | −6.3813 | [−6.5269, −6.2357] | 0.0000 |

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
