# Peer review of "Advancement in Agriculture Approaches with Agrivoltaics Natural Cooling in Large Scale Solar PV Farms"

_agriculture, doi:10.3390/agriculture13040854_

Round 1

Reviewer 1 Report

I appreciate the authors efforts on agrivoltaics natural Cooling in Large Scale Solar PV Farms. For better understanding of readers I suggest following corrections in the paper and answer the queries raised: -

1)      Line 55, page 2, word increasement is to be replaced by increased.

2)      In Table-1 and Table-2, the alignment of text may be left aligned in place of center aligned for better readability.

3)      I appreciate the authors’ style of presenting the literature review.

4)      Line 190-191, page 7, provide the geographical details for better understanding to readers.

5)      Authors’ has provided their inferences mainly on statistic grounds. I suggest them to please also mention the statistical results in terms of energy generation form.

6)      Will the higher humidity near the solar panels affect the life span of solar panels?

7)      Has position of the plots with reference to plot 7, affects the power output?

8)      Why plots 3,4,11 has more output as compared to the plot 9,12. Is it the shadow effect? Or something else?

Author Response

Thank you for the comments.

Reviewer 2 Report

The manuscript is interesting and clear. Despite the content is moderately addressing a basic research aspect, it is very practical and can be interesting for the journal readers

Author Response

Thank you for the support

Reviewer 3 Report

This manuscript shares both active and passive cooling approach in solar PV applications with emphasis on the newly develop agrivoltaic natural cooling. This manuscript also has shown a complete systematic story with reasonable conclusions.The supplementary data is scientific and rigorous.

Author Response

Thank you for the support.
